# Control of self-assembly pathways toward conglomerate and racemic supramolecular polymers

Marius Wehner[1,2], Merle Insa Silja Röhr [1], Vladimir Stepanenko[2] & Frank Würthner [1,2]✉

Homo- and heterochiral aggregation during crystallization of organic molecules has significance both for fundamental questions related to the origin of life as well as for the separation of homochiral compounds from their racemates in industrial processes. Herein, we analyse these phenomena at the lowest level of hierarchy – that is the self-assembly of a racemic mixture of (*R*,R)- and (*S*,S)-**PBI** into 1D supramolecular polymers. By a combination of UV/vis and NMR spectroscopy as well as atomic force microscopy, we demonstrate that homochiral aggregation of the racemic mixture leads to the formation of two types of supramolecular conglomerates under kinetic control, while under thermodynamic control heterochiral aggregation is preferred, affording a racemic supramolecular polymer. FT-IR spectroscopy and quantum-chemical calculations reveal unique packing arrangements and hydrogen-bonding patterns within these supramolecular polymers. Time-, concentration- and temperature-dependent UV/vis experiments provide further insights into the kinetic and thermodynamic control of the conglomerate and racemic supramolecular polymer formation.

[1] Center for Nanosystems Chemistry & Bavarian Polymer Institute, Universität Würzburg, Theodor-Boveri-Weg, 97074 Würzburg, Germany. [2] Institut für Organische Chemie, Universität Würzburg, Am Hubland, 97074 Würzburg, Germany. ✉email: wuerthner@uni-wuerzburg.de

Homochiral and heterochiral aggregation during crystallization of organic racemates have not only been discussed in the context of the origin of homochirality on earth[1] but are also of paramount importance for the separation and purification of racemic mixtures in pharmaceutical and agricultural industry or biotechnology[2,3]. In general, crystallization of racemic mixtures affords in over 90% of all cases racemic compounds containing both enantiomers in an ordered fashion within the same crystal, while far less frequently, conglomerates are formed that are equimolar mixtures of homochiral crystals[4,5] as first reported in the seminal work of Pasteur[6]. However, until now, the outcome of crystallization processes can still not be accurately predicted due to the complex interplay between thermodynamic and kinetic factors governing nucleation and crystal growth[5,7,8]. Hence, a reduction of the complexity from three-dimensional (3D) crystallization to one-dimensional (1D) supramolecular self-assembly processes offers an appealing alternative to acquire deeper mechanistic insight into conglomerate and racemic compound formation at its origin.

Indeed, within the past 20 years, research on self-assembly advanced significantly towards structures of increasing size including 1D and two-dimensional (2D) supramolecular polymers[9,10]. While this research relied for many years on thermodynamic control[11–14], i.e. the generation of equilibrium structures, only recently an advancement towards kinetically controlled supramolecular polymerization processes[15–21] and the smart utilization of out-of-equilibrium species took place[22–24]. Thus combining knowledge on the thermodynamics with detailed kinetic analysis enabled the control of supramolecular polymerization pathways towards certain supramolecular architectures[25–27]. Especially experimental parameters like the solvent[28–30], temperature[31,32] or external stimuli such as irradiation[33,34], stirring[35,36] and seeding[37–43] proved as viable tools to control the pathway complexity of supramolecular polymerizations that led to the discovery of a multitude of examples where different supramolecular polymorphs were formed[44–46].

An interesting example of supramolecular polymorphism has recently been reported by our group. Thus, for enantiomerically pure perylene bisimide (PBI) dye (R,R)-PBI, three polymorphic supramolecular polymers ((R,R)-Agg 1–3) could be produced at the same concentration in the same solvent by a proper choice of ultrasound conditions[46]. Accordingly, we were wondering whether racemic mixtures of (R,R)-PBI with its enantiomer (S,S)-PBI would follow the same kinetic self-assembly pathways to give conglomerates or if a new racemic supramolecular polymer could be formed under properly chosen self-assembly conditions.

While thermodynamic and kinetic characteristics of conglomerate and racemic compound formation have been studied in 3D crystallization processes[5,7,8] or 2D self-assembly processes on surfaces[47,48], mechanistic insight into 1D supramolecular conglomerate and racemic supramolecular polymer formation within a single racemic supramolecular system is still lacking.

Herein we provide a detailed mechanistic study on conglomerate versus racemic compound formation in the 1D supramolecular polymerization of a racemic mixture of (R,R)- and (S,S)-PBI. It is demonstrated that conglomerate helical supramolecular polymer formation proceeds under kinetic control leading to two different conglomerates Con-Agg 1 and Con-Agg 2, while under thermodynamic control, racemic supramolecular nanorod (Rac-Agg 4) formation by heterochiral aggregation of (R,R)- and (S,S)-PBI is preferred.

## Results

**Supramolecular synthesis and spectroscopic studies.** (R,R)- and (S,S)-PBI (Fig. 1a) were prepared following our previously

reported synthetic procedure for (R,R)-PBI (Supplementary Fig. 1)[46]. According to ultraviolet/visible (UV/vis) absorption spectroscopy (Fig. 1b, dotted lines and Supplementary Fig. 2), self-assembly of (S,S)-PBI under ultrasonication in a solvent mixture of methylcyclohexane (MCH) and toluene (Tol) at a volume ratio of 5:4 ($c_T \geq 3.0 \times 10^{-4}$ M) follows the same kinetic and thermodynamic pathways as reported for (R,R)-PBI[46], thereby providing the supramolecular polymorphs ((S,S)-Agg 1–3).

However, the 1D supramolecular polymorphs (S,S)-Agg 1–3 exhibit opposite helicities than their respective enantiomeric polymorphs as evidenced by mirror image circular dichroism (CD) spectra (Supplementary Fig. 3).

For the investigation of conglomerate versus racemic compound formation (Fig. 1a), racemic mixtures of (R,R)- and (S,S)-PBI in MCH/Tol (5:4 v/v) were prepared as described in the "Methods" section. First insights into whether a supramolecular conglomerate or a racemic supramolecular polymer is formed can be gained by UV/vis absorption spectroscopy: If the racemic mixture forms a supramolecular conglomerate, that is, an equal mixture of enantiomeric supramolecular polymers of (R,R)- and (S,S)-PBI formed by homochiral aggregation of the individual enantiomers, the absorption spectrum of the conglomerate must be identical to those obtained for the enantiopure polymorphs. In contrast, formation of hetero contacts between the π scaffolds within a racemic supramolecular polymer should lead to different electronic couplings[49] between the PBI chromophores and, hence, to an absorption spectrum that differs from those observed for the enantiopure compounds.

We first investigated the racemic mixture under the identical conditions ($c_T = 3.0 \times 10^{-4}$ M in MCH/Tol (5:4 v/v) at 298 K) where (R,R)-PBI formed the three supramolecular polymorphs (R,R)-Agg 1–3[46]. Upon cooling the hot racemic mixture ($c_T = 3.0 \times 10^{-4}$ M) to 298 K, conglomerate Con-Agg 1 is formed instantaneously which is composed of (R,R)-Agg 1 and (S,S)-Agg 1 in a 1:1 ratio resulting in the same absorption spectra of Con-Agg 1 and (R,R)-Agg 1 (Fig. 1b). Con-Agg 1 has an absorption maximum located at $\lambda = 496$ nm which is 26 nm hypsochromically shifted with regard to its constituting monomers (Supplementary Fig. 4) and shows another weaker band at longer wavelengths. Such an absorption profile is typically observed for PBI H-aggregates[49] with rotationally displaced PBI molecules[50,51]. As expected for a racemic mixture, Con-Agg 1 does not show any CD signal (Fig. 1c). Atomic force microscopy (AFM) studies reveal the same nanoparticle-like morphology of Con-Agg 1 as observed for (R,R)- and (S,S)-Agg 1 (Fig. 2a, d, g and Supplementary Table 1) with a diameter of $3.8 \pm 0.3$ nm, corresponding to small 1D helical oligomers composed of hydrogen-bonded dimer pairs with an interplanar π–π distance of 3.6 Å as reported previously for (R,R)-Agg 1[46]. Further evidence for conglomerate formation can be obtained from variable temperature nuclear magnetic resonance (VT-NMR) spectroscopy in deuterated toluene, where the dimerization process of the respective monomers into (R,R)- and (S,S)-Agg 1 leading to Con-Agg 1 dimers upon cooling from 365 to 262 K can be followed (for details, see Supplementary Fig. 5 and discussion there). Interestingly, the VT-NMR spectra of the dimerization of Con-Agg 1 look identical to that of (R,R)-PBI into (R,R)-Agg 1 dimers at all investigated temperatures. This proves that Con-Agg 1 consists of homochiral (R,R)- and (S,S)-Agg 1 since hetero contact formation would result in differently arranged dimers with both different temperature-dependent NMR spectra and UV/vis-absorption characteristics in comparison to enantiopure (R,R)-Agg 1.

To test whether the racemic mixture follows the same self-assembly pathways as the enantiopure PBIs, Con-Agg 1 was first sonicated under "kinetic conditions" at 293–298 K (Fig. 1a),

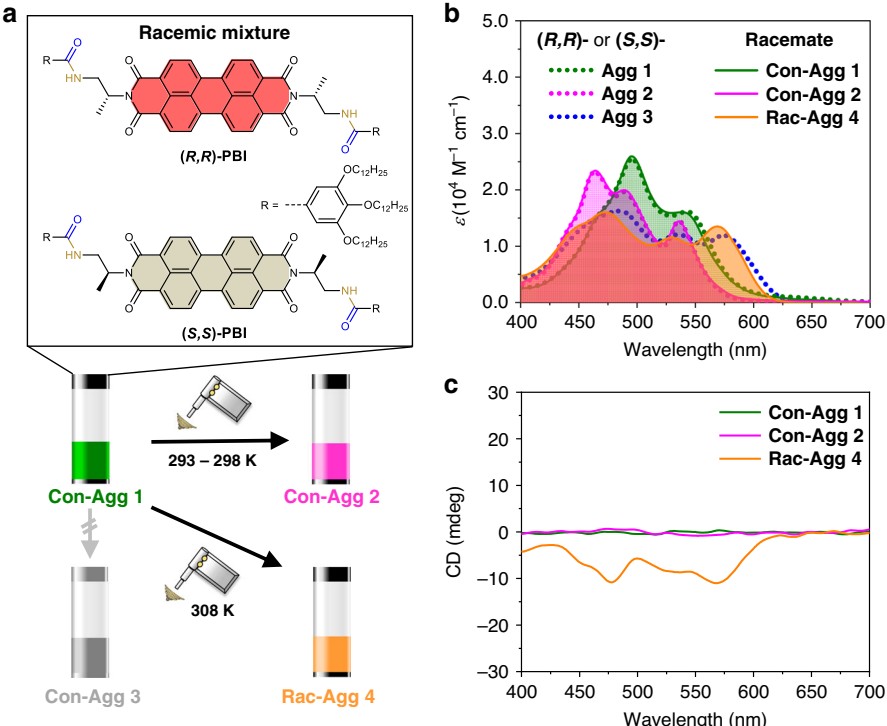

**Fig. 1 Spectroscopic studies of the racemic mixture of (R,R)- and (S,S)-PBI. a** Chemical structures of **(R,R)-** and **(S,S)-PBI** and schematic depiction of the ultrasound-induced supramolecular polymerization of the racemic mixture of **(R,R)-** and **(S,S)-PBI** into the conglomerates **Con-Agg 1** and **Con-Agg 2** and racemic supramolecular polymer **Rac-Agg 4**. **b** UV/vis-absorption and **c** CD spectra of the three different supramolecular polymorphs of the racemic mixture of **(R,R)-** and **(S,S)-PBI** (solid lines; $c_T = 3.0 \times 10^{-4}$ M for **Con-Agg 1** and **Con-Agg 2**; $c_T = 4.0 \times 10^{-4}$ M for **Rac-Agg 4**; 298 K) in MCH/Tol (5:4, v/v). For comparison, the absorption spectra of **(R,R)-Agg 1–3** (dotted lines) have been included in **b**. We note that the observed CD spectrum of **Rac-Agg 4** is an LD artefact[52] (further discussed in Supplementary Information).

which are the reported conditions for the ultrasound-induced transformation of **(R,R)-Agg 1** into the kinetic polymorph **(R,R)-Agg 2**[46]. Indeed, **Con-Agg 1** was successfully transformed into the supramolecular conglomerate **Con-Agg 2** as evidenced by the identical absorption spectra of **Con-Agg 2** and **(R,R)-Agg 2** (Fig. 1b). Compared to **Con-Agg 1**, the absorption maximum of **Con-Agg 2** is further hypsochromically shifted to $\lambda = 464$ nm. Additional absorption bands appear at $\lambda = 489$ and 536 nm. The latter can be ascribed to the characteristic J-type excitonic coupling upon rotational displacement of the PBI dyes[38,51]. As expected, **Con-Agg 2** does not show any CD signal (Fig. 1c).

Conglomerate formation becomes directly visible by investigating the morphology of **Con-Agg 2** by AFM (Fig. 2b and Supplementary Fig. 6 for enlarged AFM images). While **(R,R)-Agg 2** and **(S,S)-Agg 2** show homochiral P- and M-helical 1D nanofibres (Fig. 2e, h), respectively, **Con-Agg 2** reveals a 1:1 mixture of exactly these 1D supramolecular polymorphs. The P- and M-helices of **Con-Agg 2** both have a helical pitch of $5.7 \pm 0.2$ nm and a diameter of $4.5 \pm 0.2$ nm, which are the same characteristics as observed for enantiopure **(R,R)-Agg 2** and **(S,S)-Agg 2** within the given error range (Supplementary Table 1). Hence, it can be concluded that upon ultrasonication at 293–298 K, **(R,R)-** and **(S,S)-Agg 1** individually transform into the homochiral nanofibres **(R,R)-** and **(S,S)-Agg 2** resulting in **Con-Agg 2**. Furthermore, homochiral aggregation was also observed for a mixture of **(R,R)-** and **(S,S)-PBI** with an enantiomeric excess (ee) of **(R,R)-PBI** of 50% at the same concentration ($c_T = 3.0 \times 10^{-4}$ M; for further details, see Supplementary Fig. 8a).

Next, it was attempted to transform **Con-Agg 1** into its thermodynamically stable state. In previous studies on **(R,R)-PBI**,

it was shown that ultrasonication of a **(R,R)-Agg 1** solution in MCH/Tol 5:4 ($c_T = 3.0 \times 10^{-4}$ M) at 308 K for 180 min resulted in the formation of the thermodynamically stable **(R,R)-Agg 3** (Fig. 1b) and that this transformation process was faster the higher the concentration, thereby indicating an on-pathway self-assembly process from the original dimer species[46]. While at a concentration of $c_T = 3.0 \times 10^{-4}$ M **Con-Agg 1** did not transform into any new supramolecular polymorph within 180 min, ultrasonication of **Con-Agg 1** at an increased concentration of $c_T \geq 4.0 \times 10^{-4}$ M at 308 K led to the formation of a new supramolecular polymorph (Fig. 1a). Unexpectedly, **Con-Agg 1** was not transformed into the conglomerate **Con-Agg 3** but rather into a new racemic supramolecular polymer termed as **Rac-Agg 4**. In contrast to the conglomerates, **Rac-Agg 4** shows a broad and unstructured absorption spectrum with three absorption maxima that resembles those of **(R,R)-** or **(S,S)-Agg 3** (Fig. 1b). In contrast to **(R,R)-** or **(S,S)-Agg 3**, however, the absorption maximum of **Rac-Agg 4** ($\lambda = 472$ nm) is blueshifted by 13 nm. The characteristic bathochromically shifted absorption band of **Rac-Agg 4** that results according to our theoretical calculations (vide infra) from longitudinally and transversally shifted PBI chromophores also appears at lower wavelengths ($\lambda = 569$ nm) and shows an extinction coefficient of $\varepsilon = 13,500$ M$^{-1}$ cm$^{-1}$, which is increased by ~1500 M$^{-1}$ cm$^{-1}$ compared to **(R,R)-** or **(S,S)-Agg 3**. Recent theoretical studies by Spano and co-workers demonstrated that, for such a slipped packing arrangement of PBIs, the large exciton band width of the absorption band might result from a rather strong coupling between Frenkel and charge transfer-mediated excitons[49]. Notably, the formation of **Rac-Agg 4** was also observed within a mixture of **(R,R)-** and **(S,S)-PBI** with an ee

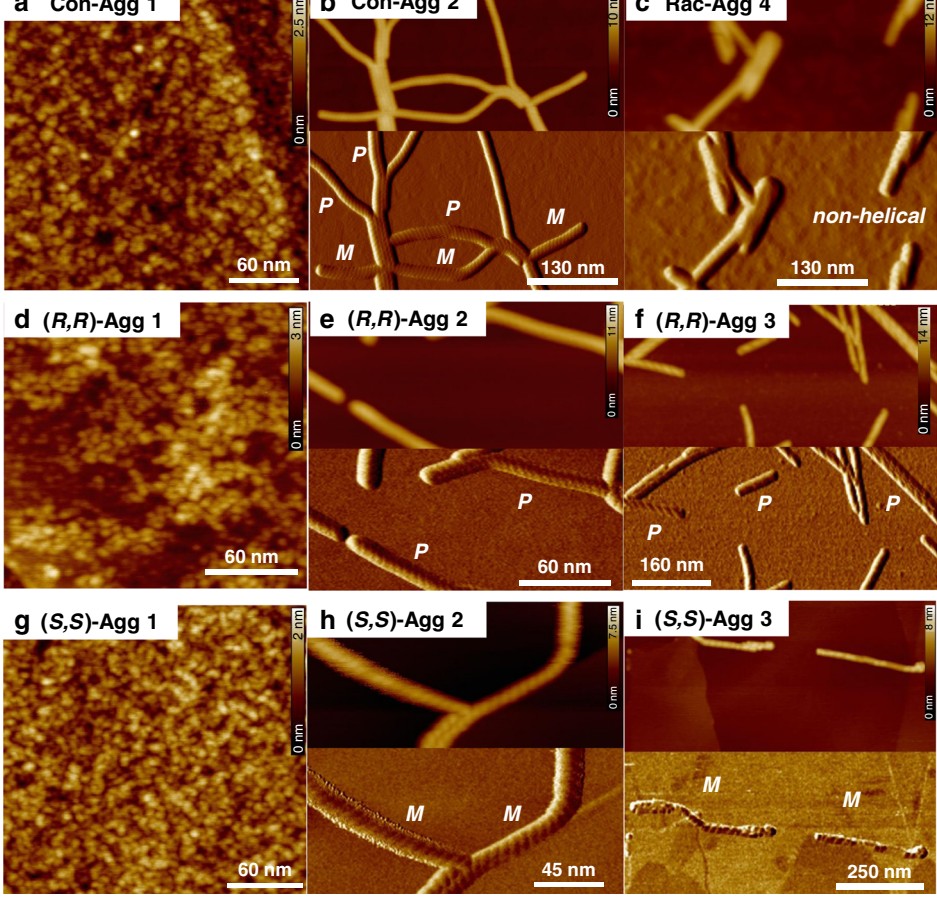

**Fig. 2 AFM studies.** AFM images of **Con-Agg 1** (**a**), **Con-Agg 2** (**b**), **Rac-Agg 4** (**c**), **(R,R)**-Agg 1–3 (**d–f**) and **(S,S)**-Agg 1–3 (**g–i**) spin-coated on HOPG from MCH/Tol (5:4, *v/v*). Z scales are 2.5 (**a**), 10 (**b**), 12 (**c**), 3 (**d**), 11 (**e**), 14 (**f**), 2 (**g**), 7.5 (**h**), and 8 nm (**i**), respectively. For **Con-Agg 1**, **(R,R)-Agg 1**, and **(S,S)-Agg 1**, only the height images are shown while for the other species both height and phase images are provided. The helicities (*P*: right-handed, *M*: left-handed) of the nanofibres of **Con-Agg 2** as well as **(R,R)-** and **(S,S)-Agg 2** and **-Agg 3** are indicated in the respective phase images.

of **(R,R)**-PBI of 50% at the same concentration ($c_T = 4.0 \times 10^{-4}$ M; for further details, see Supplementary Fig. 8b).

In contrast to **Con-Agg 1** and **Con-Agg 2**, **Rac-Agg 4** showed a negative monosignate CD signal with two minima at $\lambda = 478$ and 568 nm (Fig. 1c), which was rather unexpected since racemic mixtures should not exhibit any CD signal. Notably, the same CD signal can be reproduced at different concentrations and is not a result of small ees of **(R,R)**- or **(S,S)**-PBI within the **Rac-Agg 4** solution (Supplementary Fig. 9). Instead, in accordance with previous reports[52,53], we noted a (partial) alignment of supramolecular nanofibres in the cuvette, leading to the contamination of the CD spectra by artefacts from linear dichroism (LD) due to the optical imperfections of our set-up[54] (further discussed in Supplementary Information and Supplementary Fig. 10). AFM studies reveal that **Rac-Agg 4** consists of short 1D nanorods (Fig. 2c) with average lengths between 40 and 55 nm and a diameter of $4.1 \pm 0.2$ nm (Supplementary Table 1) that further agglomerate into sheet-like structures (Supplementary Fig. 7), which is in line with the observed LD effect. Most notably, the non-helical nanorods of **Rac-Agg 4** constitute a unique structure that has never been observed during the supramolecular polymerization of enantiopure **(R,R)**- and **(S,S)**-PBI (Fig. 2d–i). Hence, it can be concluded that those non-helical nanorods result from the formation of hetero contacts between **(R,R)**- and **(S,S)**-PBI molecules within **Rac-Agg 4**.

For the elucidation of the hydrogen-bonding pattern within the racemic supramolecular polymorph **Rac-Agg 4**, Fourier-transform

infrared (FT-IR) spectroscopy was conducted, which revealed N–H stretching frequencies at 3415, 3397 and 3306 cm$^{-1}$ (Supplementary Fig. 11). According to our previous studies[46], the peaks at 3415 and 3397 cm$^{-1}$ may be attributed to stretching vibrations of N–H groups incorporated into intermolecular hydrogen bonds between amide protons and imide oxygens of adjacent PBI molecules, which have only been observed for **(R,R)-Agg 1**. The fact that two distinct frequencies are observed may be explained by two different arrangements within the hydrogen-bonded supramolecular polymer strand, i.e. between the same enantiomers and between different enantiomers. The broad peak at 3306 cm$^{-1}$ is characteristic for intermolecular amide–amide hydrogen bonds and has also been observed for **(R,R)-Agg 3**[46]. Due to the broadness of this signal, the two types of intermolecular amide–amide hydrogen bonds between homochiral and heterochiral contacts presumably overlap. Since two different types of intermolecular amide–imide hydrogen bonds are observed for **Rac-Agg 4**, a structure with alternating monomers $[\bullet R \bullet\bullet S \bullet\bullet R \bullet\bullet S\bullet]_n$ (R and S denote **(R,R)**- and **(S,S)**-PBI, respectively) can be ruled out since in such a structure only one type of intermolecular amide–imide hydrogen bond should be observable. Instead, the present data in combination with mechanistic insights from kinetic studies (see below) indicate a structure of **Rac-Agg 4** in which **(R,R)**- and **(S,S)**-PBI dimers are connected in an alternate fashion ($[\bullet R \bullet\bullet R \bullet\bullet S \bullet\bullet S\bullet]_n$) with intermolecular amide–amide hydrogen bonds on one side and intermolecular amide–imide hydrogen bonds on the other side.

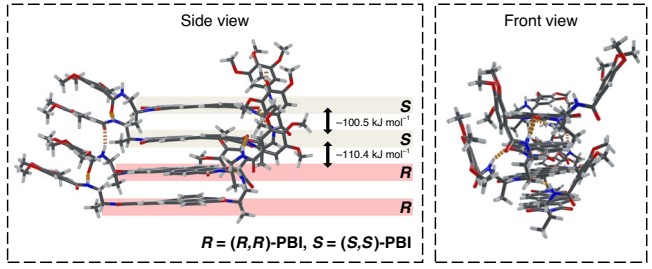

**Fig. 3 Quantum chemical calculations.** Side and front view of a geometry-optimized tetramer structure extracted from an octameric stack of **Rac-Agg 4** (Supplementary Fig. 13). Hydrogen bonds are indicated by orange dashes and dodecyloxy chains have been replaced by methoxy groups.

It is noteworthy that all three polymorphs of the racemic mixture can also be prepared at the same concentration of $c_T = 5.0 \times 10^{-4}$ M by the above-mentioned ultrasonication procedure (Fig. 1a). They are kinetically stable and can be isolated in the solid state, which justifies the designation of **Con-Agg 1** and **Con-Agg 2** as supramolecular polymer conglomerates and **Rac-Agg 4** as a racemic supramolecular polymer (Supplementary Fig. 12).

**Theoretical calculations**. To get further insight into the structure of **Rac-Agg 4**, quantum chemical calculations were performed for an octameric structural model of **Rac-Agg 4**, consisting of (**R,R**)- and (**S,S**)-PBI dimers organized in alternating order. The subsequent optimization using PM6-D3H4[55–57] correction as implemented in the MOPAC software package[58] gives rise to a relaxed octameric structure (Supplementary Fig. 13) from which, for better clarity, a tetramer section is extracted and shown in Fig. 3.

Within **Rac-Agg 4**, the PBI molecules show a $\pi$–$\pi$ stacking distance of 3.4 Å, average longitudinal displacements of 0.7 and 2.1 Å within the homo and hetero contacts, respectively, and no transversal displacement. The respective PBI molecules within the homodimer and heterodimer show average rotational displacements of 30° and 24°, respectively. Notably, the PBI molecules within the *RR*- and the *SS*-dimers as well as those within the *RS*- and *SR*-dimers are rotated in opposite directions ("enantiomeric dimers"). Hence, the rotational displacements "cancel out" over the whole octameric stack resulting in a non-helical strand, which is in accordance with our AFM results (Fig. 2c). The geometry-optimized structure of **Rac-Agg 4** shows intermolecular amide–amide hydrogen bonds on one side and intermolecular amide–imide hydrogen bonds on the other side (Fig. 3). In order to estimate the "stabilization energy" upon packing of (**R,R**)- and (**S,S**)-PBI into a racemic supramolecular polymorph, we employed single point calculations on the isolated monomer and the homochiral (*RR* or *SS*) and heterochiral (*RS* or *SR*) dimers within the stack. Subtracting the gas phase energy of the isolated monomers from the respective dimer energies gives rise to the stabilization energy upon aggregation, which is on average $-100.5$ kJ mol$^{-1}$ for the homochiral (**R,R**)- and (**S,S**)-PBI dimers and $-110.4$ kJ mol$^{-1}$ for the heterochiral dimers. The greater energy gain upon hetero contact formation might result from additional dispersion interactions between the peripheral phenyl rings[59,60] that are in close proximity to each other (middle *RS*-dimer in the front view, Fig. 3 right). Such additional interactions are not observed within the homodimers in which the respective phenyl rings point to opposite sides (lower *RR*-dimer in the front view, Fig. 3 right). Therefore, heterochiral aggregation is significantly more favourable and the main driving force for the generation of the thermodynamically stable state **Rac-Agg 4**.

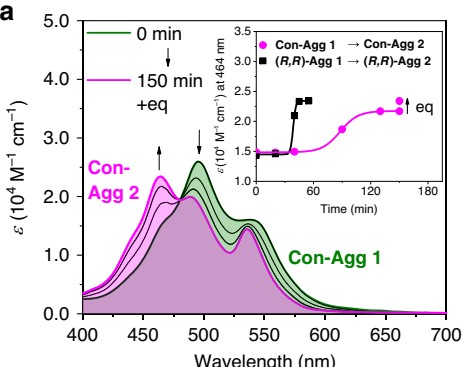

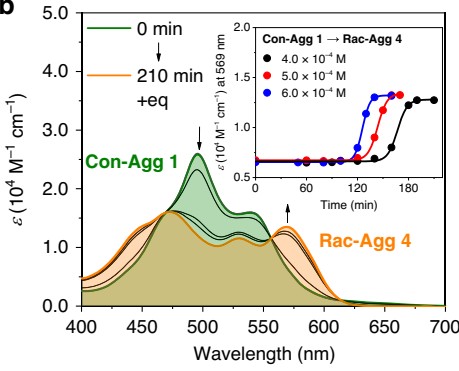

**Fig. 4 Kinetic analyses of conglomerate and racemic supramolecular polymer formation. a** UV/vis absorption spectra for the transformations of **a Con-Agg 1 → Con-Agg 2** ($c_T = 3.0 \times 10^{-4}$ M, sonication at 293–298 K) and **b Con-Agg 1 → Rac-Agg 4** ($c_T = 4.0 \times 10^{-4}$ M, sonication at 308 K) in MCH/Tol (5:4, *v/v*) depending on the sonication period. Inset of **a**: Comparison of the $\varepsilon$ values at $\lambda = 464$ nm of the ultrasound-induced transformations **Con-Agg 1 → Con-Agg 2** with those of the transformation of (**R,R**)-**Agg 1 → (R,R)-Agg 2** ($c_T = 3.0 \times 10^{-4}$ M). Inset of **b**: Plot of the time-dependent $\varepsilon$ values at $\lambda = 569$ nm at different concentrations for the ultrasound-induced transformation of **Con-Agg 1 → Rac-Agg 4**. "Eq" denotes equilibration achieved by letting the solution stand at room temperature after complete transformation. Lines are set to guide the eye.

**Kinetic analyses**. In order to gain further insight into the conglomerate and racemic supramolecular polymer formation, we performed time- and concentration-dependent UV/vis spectroscopic studies of the ultrasound-induced transformation of **Con-Agg 1** into **Con-Agg 2** or **Rac-Agg 4** (Fig. 4). The UV/vis spectra (Fig. 4a) and plot of $\varepsilon$ values (inset Fig. 4a) at the absorption maximum of **Con-Agg 2** ($\lambda = 464$ nm) show a gradual transformation of **Con-Agg 1** into **Con-Agg 2** upon ultrasonication at 293–298 K. While at a concentration of $c_T = 3.0 \times 10^{-4}$ M this process is completed within 120 min, it takes only 70 min at $c_T = 6.0 \times 10^{-4}$ M (Supplementary Fig. 14), which indicates that **Con-Agg 2** is formed in an on-pathway fashion, i.e. **Con-Agg 1** directly transforms into **Con-Agg 2** without dissociation into monomers as intermediates[25,26]. The same behaviour has also been observed for the self-assembly of enantiomerically pure (**R,R**)-**Agg 1** into (**R,R**)-**Agg 2**[46]. Notably, clear isosbestic points can be observed in the UV/vis spectra indicating that only **Con-Agg 1** and **Con-Agg 2** are present during the transformation process and no **Rac-Agg 4** is formed under these "kinetic" ultrasonication conditions. The transformation of **Con-Agg 1** into **Con-Agg 2** is >60 min slower compared to the transformation of enantiopure (**R,R**)-**Agg 1** into (**R,R**)-**Agg 2** under the same experimental conditions (inset Fig. 4a), which fits well with conglomerate formation because in a **Con-Agg 1** solution with a concentration

of $c_T = 3.0 \times 10^{-4}$ M the individual concentrations of (**R,R**)- and (**S,S**)-**PBI** are $1.5 \times 10^{-4}$ M, and hence, the on-pathway transformation—a process that becomes slower upon lowering the concentration—of **Con-Agg 1** into **Con-Agg 2** is slower compared to the transformation of (**R,R**)-**Agg 1** into (**R,R**)-**Agg 2** at the same concentration of $c_T = 3.0 \times 10^{-4}$ M.

Ultrasonication of **Con-Agg 1** at $c_T \geq 4.0 \times 10^{-4}$ M at 308 K leads to the transformation into **Rac-Agg 4** (Fig. 4b). Again, isosbestic points in the UV/vis spectra prove the existence of only two species, i.e. **Con-Agg 1** and **Rac-Agg 4**. A plot of the concentration- and time-dependent $\varepsilon$ values at $\lambda = 569$ nm reveals that the transformation of **Con-Agg 1** is faster the higher the concentration of **Con-Agg 1** (inset Fig. 4b), indicating again that **Con-Agg 1** dimer units are on-pathway intermediates to **Rac-Agg 4**. Notably, this implies that no disassembly into monomers takes place prior further aggregation into **Rac-Agg 4**, and hence, this also precludes an alternating $[\bullet R \bullet\bullet S \bullet\bullet R \bullet\bullet S \bullet]_n$ structure of **Rac-Agg 4**. Together with the fact that all transformations within the supramolecular polymorphs of enantiopure (**R,R**)-**PBI** occur on the dimer level[46], this result further supports the calculated $[\bullet R \bullet\bullet R \bullet\bullet S \bullet\bullet S \bullet]_n$ structure of **Rac-Agg 4** with alternating dimers. Furthermore, the non-linear sigmoidal kinetics of **Rac-Agg 4** formation is characteristic for an autocatalytic process following the cooperative aggregation mechanism[11,37,38,46], which is proven in the next section.

**Thermodynamic analyses.** While the thermodynamic characteristics of **Con-Agg 1** and **Con-Agg 2** can be deduced from previous studies on (**R,R**)-**Agg 1** and (**R,R**)-**Agg 2**[46] (Supplementary Table 2 and discussion there), temperature-dependent UV/vis spectroscopic studies were conducted to elucidate the aggregation mechanism and the thermodynamic parameters of the racemic supramolecular polymer **Rac-Agg 4** (Fig. 5).

Since **Rac-Agg 4** is only accessible at higher concentration under ultrasound irradiation, the thermodynamics of **Rac-Agg 4** formation cannot be monitored. However, the dissociation process of **Rac-Agg 4** can be analysed by temperature-dependent UV/vis spectroscopy. Upon heating of **Rac-Agg 4** from 293 K to 340 K (Fig. 5a), the characteristic lowest energy absorption band of **Rac-Agg 4** at $\lambda = 569$ nm and the broad absorption spectrum of **Rac-Agg 4** converts into an absorption spectrum resembling a mixture of **Con-Agg 1** and monomers (Fig. 1b and Supplementary Fig. 4). The lack of defined isosbestic points further supports the involvement of these three species. A plot of absorbance at $\lambda = 569$ nm (inset Fig. 5b) reveals a characteristic shape for a cooperative aggregation mechanism for **Rac-Agg 4**[11,13,61] and critical temperatures $T_E$ that increase from 330 K to 332 K upon increasing concentration from $c_T = 4.0 \times 10^{-4}$ to $6.0 \times 10^{-4}$ M. According to a previously reported procedure[46], we determined the overall monomer concentrations $c_{Mono}$ at the respective critical temperatures $T_E$ for different concentration of **Rac-Agg 4** from which we prepared a Van't Hoff plot giving a standard enthalpy release $\Delta H^O = -109$ kJ mol$^{-1}$ and standard entropy $\Delta S^O = -243$ J mol$^{-1}$ K$^{-1}$ upon elongation as well as an association constant of $K_E = 2.6 \times 10^6$ M$^{-1}$ and a standard Gibbs free energy of $\Delta G^O = -36.6$ kJ mol$^{-1}$ at 298 K (Supplementary Table 2).

**Potential energy surface of the supramolecular polymerization.** To construct a potential energy surface of the supramolecular conglomerate and racemic supramolecular polymer formation, we will first summarize the polymer structures and thermodynamic features of **Con-Agg 1** and **Con-Agg 2** (Fig. 6 and Supplementary Table 2). Since these two conglomerates are composed of equal mixtures of homochiral (**R,R**)- and (**S,S**)-**Agg 1** or (**R,R**)- and (**S,S**)-**Agg 2**, the aforementioned characteristics

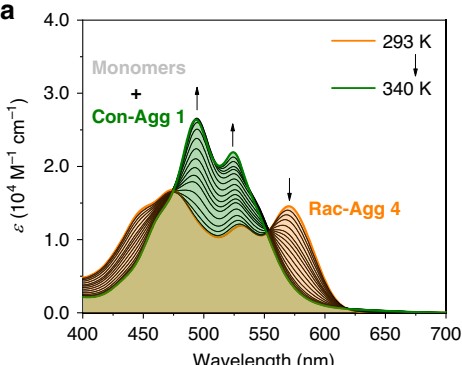

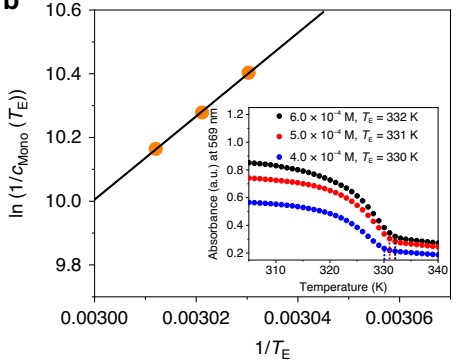

**Fig. 5 Thermodynamic analysis of Rac-Agg 4. a** Temperature-dependent UV/vis absorption spectra of **Rac-Agg 4** upon heating ($c_T = 4.0 \times 10^{-4}$ M, $T = 293$ K → 340 K, heating rate: 1 K min$^{-1}$). **b** Van't Hoff analysis of the thermodynamic parameters of the elongation of **Rac-Agg 4**. Inset: Plot of the absorbances at $\lambda = 569$ nm as a function of temperature and concentration. The dotted lines indicate the respective elongation temperatures.

are known from our previous extensive studies on (**R,R**)-**PBI**[46]. Thus **Con-Agg 1** consists of tightly bound homochiral $M$- and $P$-helical (**R,R**)- and (**S,S**)-**Agg 1** dimers (Supplementary Fig. 3), which are only loosely connected to each other by small elongation constants in an anti-cooperative aggregation mechanism[50]. Within those dimers, the respective PBI dyes are connected by a combination of π–π interactions and intermolecular amide–imide hydrogen bonds that show N–H stretching frequencies of 3406 and 3377 cm$^{-1}$ while only weak van-der-Waals interactions are given between the respective dimers.

In contrast, **Con-Agg 2** is formed by a cooperative supramolecular polymerization and shows helical structures in which the respective PBI monomers within the homochiral $P$- and $M$-helical (**R,R**)- and (**S,S**)-**Agg 2** are connected by π–π interactions and intermolecular amide–amide hydrogen bonds on one side (N–H stretching frequency at 3336 cm$^{-1}$) and intramolecular amide–imide hydrogen bonds on the other side (N–H stretching frequencies at 3406 cm$^{-1}$).

From the thermodynamic studies, it becomes obvious that racemic supramolecular **Rac-Agg 4** is the thermodynamically stable state showing the highest Gibbs free energy release upon elongation with $\Delta G^O = -36.6$ kJ mol$^{-1}$ while **Con-Agg 1** shows $\Delta G^O$ values of $-35.9$ kJ mol$^{-1}$ for its dimerization and $-30.0$ kJ mol$^{-1}$ for further elongation of these dimers and **Con-Agg 2** a value of $\Delta G^O$ (298 K) = $-33.5$ kJ mol$^{-1}$. Thus, upon increasing aggregate size, the thermodynamic stabilities follow the trend $\Delta G^O$ (**Con-Agg 1**) > $\Delta G^O$ (**Con-Agg 2**) > $\Delta G^O$ (**Rac-Agg 4**), meaning that **Rac-Agg 4** is the most stable and **Con-Agg 1** the least stable species (Fig. 6 and Supplementary Fig. 15). This can

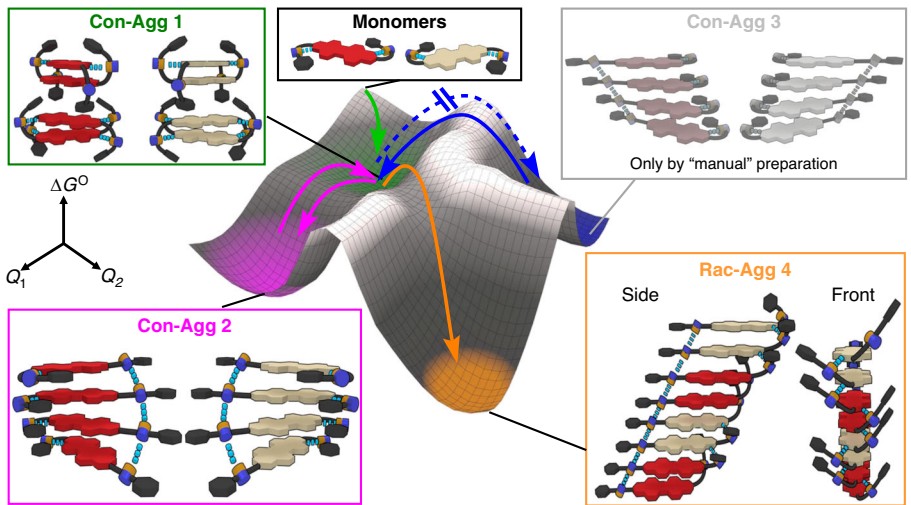

**Fig. 6 Qualitative energy landscape.** Qualitative energy for the supramolecular conglomerate and the racemic supramolecular polymer formation from the racemic mixture of **(R,R)-** and **(S,S)-PBI**. The aggregate structures corresponding to supramolecular pathways, which are directly accessible by ultrasonication of the racemic mixture (**Con-Agg 1**, **Con-Agg 2** and **Rac-Agg 4**), are schematically illustrated in colour while the structure of the polymorph **Con-Agg 3**, which can only be prepared by manual mixing of prefabricated **(R,R)-** and **(S,S)-Agg 3**, is shown in grey. The colour code of the respective PBIs is adjusted to Fig. 1a; hydrogen bonding is indicated with blue dashes. $Q_1$ and $Q_2$ denote coordinates for the lowest-energy reaction pathway between the different polymorphs.

be explained by the formation of energetically favourable heterochiral contacts between **(R,R)-** and **(S,S)-PBI** dimers (Fig. 3) but is also a result of stronger hydrogen bonding within **Rac-Agg 4** (Supplementary Fig. 11) compared to that of **Con-Agg 1** and **Con-Agg 2**[46]. Especially the intermolecular amide–amide hydrogen bonds within **Rac-Agg 4** with a N–H stretching frequency of 3306 cm$^{-1}$ are significantly stronger (indicated by a lower wavenumber) than the respective amide–amide hydrogen bonds in **Con-Agg 2** (3336 cm$^{-1}$) or than the intermolecular amide–imide hydrogen bonds within **Con-Agg 1** (3406 and 3377 cm$^{-1}$). Thus the hydrogen-bonding pattern of **Rac-Agg 4** results in the energetically most favourable and most ordered structure with the highest standard enthalpy release and the lowest (most unfavourable) standard entropy upon aggregation (Supplementary Table 2). **Rac-Agg 4** does not spontaneously transform into any other species at room temperature over time and additional experiments to transform **Rac-Agg 4** into another species upon prolonged (>2 h) ultrasonication ($T \le 313$ K) failed, which reflects the high thermodynamic stability of **Rac-Agg 4**.

Based on these data, the potential energy surface shown in Fig. 6 was constructed that reflects all the thermodynamic features of the supramolecular polymerization of the racemic mixture of **(R,R)-** and **(S,S)-PBI** into the supramolecular conglomerates **Con-Agg 1** and **Con-Agg 2** and the racemic supramolecular polymer **Rac-Agg 4**, respectively. Furthermore the energy surface includes the polymerization pathway to **Con-Agg 3**—a 1:1 mixture of **(R,R)-** and **(S,S)-Agg 3**—which is not directly formed by ultrasonication of the racemic mixture (blue dashed arrow, Fig. 6), although **(R,R)-** and **(S,S)-Agg 3** have been previously observed for the pure enantiomers[46] (see discussion below).

On a qualitative level, Fig. 6 also includes the transition energies of the self-assembly pathways deduced from our kinetic data. Thus, upon cooling of a concentrated ($c_T \ge 3.0 \times 10^{-4}$ M) racemic mixture of **(R,R)-** and **(S,S)-PBI**, **Con-Agg 1** is formed instantaneously (green arrow, Fig. 6), which is a kinetic conglomerate consisting of equal amounts of small **(R,R)-** and **(S,S)-Agg 1** oligomers containing strongly bound dimeric species (Supplementary Fig. 5). **Con-Agg 1** is the central intermediate

towards the formation of conglomerate **Con-Agg 2** and racemic compound **Rac-Agg 4**. Under kinetic conditions, i.e. ultrasonication at 293–298 K, **Con-Agg 1** is transformed into the more stable **Con-Agg 2** (magenta arrow, Fig. 6), which is a kinetic polymorph. The kinetic barrier between **Con-Agg 1** and **Con-Agg 2** that has to be overcome is lower compared to that between **Con-Agg 1** and **Rac-Agg 4**, and hence, **Con-Agg 2** forms at lower temperatures of 293–298 K upon ultrasonication although it is less thermodynamically stable than **Rac-Agg 4**.

Transformation of **Con-Agg 1** under thermodynamic conditions (ultrasonication at 308 K) allows the system to transform into the thermodynamically most stable state **Rac-Agg 4** (orange arrow, Fig. 6). This reveals the most striking difference to the supramolecular polymerization of the single enantiomer systems, **(R,R)-** or **(S,S)-PBI**, leading to **(R,R)-** or **(S,S)-Agg 3**, respectively. The inaccessibility of **Con-Agg 3** upon ultrasonication of the racemic mixture as described in the "Methods" section (see below) was rather unexpected since **(R,R)-Agg 3** was reported to be the thermodynamically stable state of the supramolecular polymerization of **(R,R)-PBI** under the same experimental conditions. The most obvious explanation is that **Rac-Agg 4** with the $[\bullet R \bullet\bullet R \bullet\bullet S \bullet\bullet S \bullet]_n$ sequence is by 2.3 kJ mol$^{-1}$ more stable than **Con-Agg 3** consisting of $[\bullet S \bullet\bullet S \bullet\bullet S \bullet\bullet S \bullet]_n$ and $[\bullet R \bullet\bullet R \bullet\bullet R \bullet\bullet R \bullet]_n$ sequences (Supplementary Table 2). Nevertheless, **Con-Agg 3** can be manually prepared from enantiopure samples of **(R,R)-** and **(S,S)-Agg 3**, respectively, Therefore, as previously reported[46], enantiopure solutions of **(R,R)-Agg 1** or **(S,S)-Agg 1** were sonicated at 308 K and transformed into **(R,R)-** or **(S,S)-Agg 3**, respectively, and afterwards mixed in a 1:1 ratio. In order to verify that the racemic supramolecular polymer **Rac-Agg 4** is the thermodynamically stable state of the racemic mixture and, hence, more stable than **Con-Agg 3**, we sonicated a solution of **Con-Agg 3** at 308 K and at 313 K (Supplementary Fig. 16). After sonication at 313 K for 6 h, **Con-Agg 3** was completely transformed into **Rac-Agg 4** proving that **Rac-Agg 4** is the thermodynamically most stable polymorph of the racemic mixture. As discussed in Supplementary Information, the polymerization pathway of manually prepared **Con-Agg 3** into **Rac-Agg 4** comprises the disassembly of **Con-Agg 3** into

**Con-Agg 1** dimers (solid blue arrow, Fig. 6) and simultaneous transformation of these intermediates into **Rac-Agg 4** (orange arrow, Fig. 6).

Additionally, we were able to transform **Con-Agg 2** into **Rac-Agg 4** upon ultrasonication at 308 K (Supplementary Fig. 17), which proceeds via disassembly of **Con-Agg 2** into **Con-Agg 1** (magenta arrow, Fig. 6) followed by the transformation of **Con-Agg 1** into **Rac-Agg 4** (orange arrow, Fig. 6). Thus the thermodynamically stable racemic supramolecular polymer **Rac-Agg 4** could be prepared from all kinetically metastable conglomerate species **Con-Agg 1–3** under appropriate ultrasonication conditions, and therefore, full pathway control of the supramolecular polymerization of the racemic mixture was achieved.

## Discussion

Understanding self-assembly pathways leading either towards conglomerates or racemic compounds by homochiral and heterochiral aggregation of organic molecules is of major importance not only from an academic but also from an industrial point of view. Herein, we described an in-depth mechanistic study of such homochiral and heterochiral aggregation in 1D self-assembly by investigating the supramolecular polymerization of the racemic mixture of (**R,R**)- and (**S,S**)-**PBI**. Precise control of the self-assembly pathways by ultrasonication led to the production of three different polymorphs, namely, two kinetically formed conglomerates **Con-Agg 1** and **Con-Agg 2** and the thermodynamically most stable racemic supramolecular polymer **Rac-Agg 4** in the same solvent mixture at the same concentration. While the conglomerates **Con-Agg 1** and **Con-Agg 2** are 1:1 mixtures of homochiral (**R,R**)- and (**S,S**)-**Agg 1** and -**Agg 2**, respectively, that are also accessible from the single enantiomers, a new supramolecular polymerization pathway was launched for the racemic mixture to afford the thermodynamically most stable racemic supramolecular polymer **Rac-Agg 4** with a unique structure of alternating (**R,R**)- and (**S,S**) homochiral dye pairs. In contrast to the enantiopure (**R,R**)-**PBI** system for which (**R,R**)-**Agg 3** was reported to be the thermodynamically most stable species[46], the polymerization pathway leading to **Con-Agg 3** was replaced for the racemic mixture by the self-assembly into thermodynamically stable racemic supramolecular polymer **Rac-Agg 4**.

With our study, we provide a demonstration of kinetic and thermodynamic pathway control towards conglomerate and racemic supramolecular polymers, thereby elucidating an important process of chiral discrimination at its origin. As obvious from our study, the in-depth elaboration of energy landscapes in self-assembly processes as provided in Fig. 6 can guide research towards the construction of sophisticated supramolecular materials. Such studies may ultimately also improve our understanding of symmetry breaking events in self-assembly processes that are not only vital for the synthesis and enantiomeric resolution of industry-relevant compounds but also explain the origin of homochirality in nature.

## Methods

**UV/vis absorption spectroscopy.** The spectroscopic measurements were conducted under ambient conditions using dry solvents of spectroscopic grade. The UV/vis spectra of the samples were measured with Jasco V-670 and Jasco V-770 spectrophotometers equipped with a PAC-743R Auto Peltier 6/8-cell changer for temperature control. The temperature-dependent absorption spectra were density corrected for the different temperatures. Solid state UV/vis spectra were recorded with a Jasco J-810 spectropolarimeter.

**CD and LD spectroscopy.** CD and LD spectra were measured with a Jasco J-810 spectropolarimeter equipped with a Jasco CDF-426S Peltier temperature controller. All CD and LD spectra were density corrected for different temperatures.

**FT-IR spectroscopy.** The FT-IR spectroscopic analyses were performed on a Jasco FT/IR-4600 spectrometer. For measurements in the solution, the spectrometer was equipped with a Falcon Mid-IR transmission accessory (temperature control module, liquid recirculator) with a fixed path cuvette with CaF$_2$ windows and path length of 1 mm (PIKE technologies).

**Atomic force microscopy.** AFM measurements were performed under ambient conditions using a Bruker Multimode 8 SPM system operating in tapping mode in air. Silica cantilevers (OMCL-AC160TS, Olympus) with a resonance frequency of ~300 kHz and a spring constant of ~40 Nm$^{-1}$ were used. The samples were spin-coated from solution onto highly ordered pyrolytic graphite with 4000 or 10,000 rpm.

**Preparation of the racemic mixture (yielding Con-Agg 1).** In order to prepare a racemic mixture of (**R,R**)- and (**S,S**)-**PBI** ([(**R,R**)-**PBI**]/[(**S,S**)-**PBI**] = 1:1), stock solutions with same concentrations were prepared for both compounds individually. Therefore, a weighed amount of a MCH and Tol mixture with a volume ratio of $v/v = 5:4$ was added to the solid sample of the respective enantiomer in a screw cap vial. After preparation of the stock solution of each enantiomer, equal volumes of (**R,R**)- and (**S,S**)-**PBI** solution were added to a new vial with a glass pipette (solution were weighed). After heating to >363 K (leading to dissociation into monomers) and cooling to 298 K in a sealed vial, the racemic nature of the solution was double checked by careful CD measurements (and potentially adjusted by addition of tiny amounts of the "minor" enantiomer solution) to ensure that equal amounts of both enantiomers are present in the racemic mixture resulting in no observable CD signal. Prior further experiments, heating to >363 K and cooling to 298 K was repeated for the racemic mixture in a sealed vial resulting in the formation of **Con-Agg 1**.

**Ultrasonication.** Ultrasonication was performed with an ultrasonic processor UP50H (30 kHz, 50 Watt, 100% amplitude) from Hielscher. For the preparation of **Con-Agg 2** or **Rac-Agg 4**, a portion of 3.1 mL of **Con-Agg 1** solution in a mixture of MCH/Tol in a volume ratio of $v/v = 5:4$ was placed in a vial and heated to >363 K leading to dissociation into monomers. The ultrasonic finger was dipped about half-way in the hot solution and the vial was sealed with a cap and parafilm. The temperature, at which ultrasonication was conducted (293–298 K or 308 K for the preparation of **Con-Agg 2** or **Rac-Agg 4**, respectively), was controlled by a water bath in which the vial was placed.

**Computational details.** The optimized structure of octameric **Rac-Agg 4** was calculated in the framework of the semi-empirical PM6 method[55] along with the D3H4[56,57] correction for an adequate description of hydrogen bonds and dispersion by using the MOPAC2016 program suite[58]. In order to decrease the computational demand, we replaced the long alkyl chains by methyl groups.

## Data availability

The authors declare that all data supporting the findings of this study are available within the article and its Supplementary Information file. Extra data are available from the corresponding author upon request.

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

## Acknowledgements

We thank Tabea Gerlach for the synthesis of some of the utilized precursor molecules. M.W. thanks the Fonds der Chemischen Industrie for a fellowship. Financial support by the State of Bavaria for the establishment of the KeyLab for Supramolecular Polymers of the Bavarian Polymer Institute (BPI) is gratefully acknowledged. This publication was supported by the Open Access Publication Fund of the University of Wuerzburg.

## Author contributions

F.W. conceived and supervised the project. M.W. did the synthesis and spectroscopic studies of the investigated molecules and their racemic mixture. M.I.S.R. provided the quantum-chemical calculation. V.S. did the AFM measurements. M.W. and F.W. co-wrote the manuscript.

## Funding

## Competing interests

The authors declare no competing interests.
