## [Peer Review File · Nature Communications]

REVIEWER COMMENTS

Reviewer #1 (Remarks to the Author):

The manuscript by Wehner et al. reports precise control of the self-assembly pathways (kinetic vs thermodynamic) by ultrasonication under different conditions, leading to the formation of three different chiral polymorphs (conglomerate or racemic supramolecular polymers). The FT-IR, NMR spectroscopy, kinetic and thermodynamic analysis based on UV/vis spectroscopic studies, as well as high resolution atomic force microscopy, and quantum-chemical calculations thoroughly supported the authors' conclusion in this paper. Although the self-assembly behaviors of enantiomerically pure (R,R)-PBI molecule have been reported previously (J. Am. Chem. Soc. 2019, 141, 6092) and many chiral supramolecular polymers with formation of conglomerate and racemic co-assembly have been reported, controlling the formation of conglomerate or racemic supramolecular polymer are still intriguing point in this work, which is clearly described and justified in the introduction part. Indeed, while the X-ray crystal analysis is sacrificed, the 1D supramolecular assembly has provided complementary and important mechanistic insight into the crystallization of chiral molecules. I find it interesting that racemic supramolecular polymers tended to agglomerate and resulted in thermodynamically stable structure, which would be suggestive in terms of the fact that 90% of racemic mixtures result in racemic compounds in the 3D crystallization. The paper reads very well, and I strongly recommend the publication of this manuscript in Nature Communications.

Minor comments:

- From the VT-NMR spectra in Supplementary Figure 4, the authors concluded the formation of Con-Agg1 consisting of homochiral (R,R)-Agg1 and (S,S)-Agg1 rather than hetero based on the identical spectral change between Con-Agg1 and (R,R)-Agg1. This experiment was conducted in good solvent toluene, in which only monomer/dimer (no Agg1) co-existed. This was validated in the authors' previous paper, but for readers' better understanding, I suggest the authors to add short discussion from the previous paper because the dimer plays an important role in the present pathway complexity.

- The heterochiral aggregation is more favored at the thermodynamically stable state and drives the formation of Rac-Agg4; what will happen at enantiomeric excess conditions (ee value: 50 % or -50 %)? Do the final self-assembly structures compose of Rac-Agg4+Agg3 in such cases? What if a mixture of (R,R)-Agg3 and (S,S)-Agg3 is agitated?

Reviewer #2 (Remarks to the Author):

This manuscript reports control on supramolecular polymerization of a mixture of enantiopure (R, R)- + (S, S) PBI derivatives leading to the formation of conglomerate and racemic structures depending on specific conditions. This phenomenon has been elucidated with careful AFM analysis, and UV/Vis spectroscopy of the homo-polymers and mixtures while the kinetic and thermodynamic analysis was conducted by spectroscopy studies. Experimental observation has been supported by theoretical studies which together were used to provide a clear picture on this complex supramolecular polymerization process and chirality issues. While chirality issues in supramolecular polymerization has been studied with significant detail in the recent past, most examples are limited to chirality induction and in the recent past chiral luminescence. In contrast the present study reflects a fundamental aspect which, although was known in context of organic crystals, has never been looked at in detail in context of 1D supramolecular polymers as rightly claimed by the authors. The paper has been written very well, it is remarkably free of error and should make a lasting impact in the field. The manuscript is recommended for publication after minor revision as noted below:

(a) It may be relevant to examine the supramolecular assembly of (R, R)- + (S, S) PBIs with stoichiometric imbalance. At least with one ratio other than 1:1 may be informative. Will it still produce a conglomerate and, in that case, will it lead to eventually homochiral assembly following majority rule?

(b) Do different aggregated states exhibit distinct emission spectra?

(c) Supplementary Figure 4: It is intriguing that the spectra at 295K are particularly broad and become relatively sharp again by lowering the temperature further. What may be the reason?

(d) A recent review article (Chem. Commun., 2020, 56, 6757) on controlled supramolecular assembly may be relevant in context of the discussion in the introduction.

Control of self-assembly pathways toward conglomerate and racemic supramolecular polymers

Marius Wehner, Merle Insa Silja Röhr, Vladimir Stepanenko, Frank Würthner

This article reports the discovery of new polymorphs of supramolecular polymers composed of a racemic mixture of perylene bisimide (PBI) as well as the formation pathways of those polymorphs. This also represents the first mechanistic study of one-dimensional supramolecular polymerization of a racemic mixture of (*R,R*) and (*S,S*)-PBI. It was determined that a racemic mixture of (*R,R*) and (*S,S*)-PBI initially formed homochiral dimer pairs in solution (the conglomerate particles of which are termed Con-Agg 1), but subsequent perturbation either by sonication (purported to induce kinetic self-assembly) or heating and cooling as well as sonication (purported to induce thermodynamic self-assembly) could induce the formation of conglomerate, H-bonded homochiral helices (termed Con-Agg 2) or periodic supramolecular polymers comprised of alternating homochiral pairs (termed Rac-Agg 4), respectively. The formation of these species (Con-Agg 2 and Rac-Agg 4) was determined to proceed via nucleation-elongation mechanisms, as evidenced by sigmoidal kinetic traces. Thermodynamic analysis of the dissociation of Rac-Agg 4, as well as quantum chemical calculations, showed that the formation of Rac-Agg 4 was more thermodynamically favorable than the formation of Con-Agg 2. Furthermore, the authors claim that the concentration-dependent kinetics of formation of Rac-Agg 4 from Con-Agg 1 indicate that Con-Agg 1 is an on-pathway intermediate to the formation of the putative thermodynamic product Rac-Agg 4. This study thus represents a thorough characterization of the thermodynamics of a system capable of supramolecular self-assembly and gives guidelines for pathway control within this system.

The purported structures of each of the novel polymorphs are supported by UV/vis-absorption studies, circular dichromism data, and quantum chemical calculations. These calculations also support the categorization of Rac-Agg 4 as the thermodynamically most stable state of the system and Con-Agg 2 as a kinetic product, since it is thermodynamically more stable than Con-Agg 1 but not as stable as Rac-Agg 4. The authors are careful not to overstate the thermodynamic stability of Rac-Agg 4, but occasionally use ambiguous language to describe the relative stability of each polymorph. For example, in the Discussion section, the authors claim that Rac-Agg 4 is formed under thermodynamic control, and in their qualitative energy surface (Figure 6) place a large energetic barrier between the minima corresponding to Con-Agg 2 and Rac-Agg 4. Since pathway independence is characteristic of thermodynamic equilibria for dynamic systems, dual entry point analysis experiments would bolster the authors' argument that Rac-Agg 4 is the most thermodynamically stable species. Specifically, the authors might consider sonication and thermal cycling of Con-Agg 2 to potentially generate Rac-Agg 4. This would determine whether Con-Agg 2 is a kinetic trap or an off-pathway intermediate en route to the putative thermodynamic product Rac-Agg 4. Since Rac-Agg 4 can also dissociate into Con-Agg 1 (homochiral dimers), it would also be worth determining whether or not Rac-Agg 4 can be converted into any other metastable species, particularly Con-Agg 2, by slow thermal cycling and prolonged sonication. This would better inform the authors' representation of the system's energetic landscape and elucidate other possible formation pathways for each polymorph.

There are occasional minor grammatical and spelling errors throughout the text, but it did not impact my understanding of the article.

This article demonstrates a generalizable method to map the energy landscape of a dynamic system capable of producing multiple products, which may become a very useful tool to

simply represent complicated dynamic systems. This work also uses thermodynamic modeling to elucidate purported kinetic and thermodynamic pathways in supramolecular polymerization, a task which is often overlooked in computational approaches to dynamic combinatorial chemistry. However, the work would be more complete and impactful if the relationship between each of the novel species reported was further elucidated. That is, analysis of pathway-dependence in the synthesis of Rac-Agg 4, as well as further thermodynamic characterization of Con-Agg 2 to determine whether it is a kinetic trap or an off-pathway intermediate would expand the author's claim of "control of self-assembly pathways." For these reasons I recommend publication with major revisions.

Responses to the referees' comments

Manuscript NCOMMS-20-26065: "Control of self-assembly pathways toward conglomerate and racemic supramolecular polymers"

Answer: We thank all reviewers for their careful reading of our manuscript and all their helpful advices for the improvement of our manuscript.

Reviewer 1:

The manuscript by Wehner et al. reports precise control of the self-assembly pathways (kinetic vs thermodynamic) by ultrasonication under different conditions, leading to the formation of three different chiral polymorphs (conglomerate or racemic supramolecular polymers). The FT-IR, NMR spectroscopy, kinetic and thermodynamic analysis based on UV/vis spectroscopic studies, as well as high resolution atomic force microscopy, and quantum-chemical calculations thoroughly supported the authors' conclusion in this paper. Although the self-assembly behaviors of enantiomerically pure (R,R)-PBI molecule have been reported previously (J. Am. Chem. Soc. 2019, 141, 6092) and many chiral supramolecular polymers with formation of conglomerate and racemic co-assembly have been reported, controlling the formation of conglomerate or racemic supramolecular polymer are still intriguing point in this work, which is clearly described and justified in the introduction part. Indeed, while the X-ray crystal analysis is sacrificed, the 1D supramolecular assembly has provided complementary and important mechanistic insight into the crystallization of chiral molecules. I find it interesting that racemic supramolecular polymers tended to agglomerate and resulted in thermodynamically stable structure, which would be suggestive in terms of the fact that 90% of racemic mixtures result in racemic compounds in the 3D crystallization. The paper reads very well, and I strongly recommend the publication of this manuscript in Nature Communications.

Minor comments:

From the VT-NMR spectra in Supplementary Figure 4, the authors concluded the formation of Con-Agg1 consisting of homochiral (R,R)-Agg1 and (S,S)-Agg1 rather than hetero based on the identical spectral change between Con-Agg1 and (R,R)-Agg1. This experiment was conducted in good solvent toluene, in which only monomer/dimer (no Agg1) co-existed. This was validated in the authors' previous paper, but for readers' better understanding, I suggest the authors to add short discussion from the previous paper because the dimer plays an important role in the present pathway complexity.

Answer: As suggested, we added a short discussion after the presentation of our NMR data in Supplementary Figure 4.

The heterochiral aggregation is more favored at the thermodynamically stable state and drives the formation of Rac-Agg4; what will happen at enantiomeric excess conditions (ee value: 50 % or -50 %)? Do the final self-assembly structures compose of Rac-Agg4+Agg3 in such cases?

Answer: We sonicated a mixture of (R,R)- and (S,S)-**Agg 1** ($c_T = 4.0 \times 10^{-4}$ M) with an ee of 50% of (R,R)-**PBI** (the solution consists of 75% (R,R)-**PBI** and 25% (S,S)-**PBI**) at 308 K. Under these conditions the racemic mixture forms **Rac-Agg 4** while an enantiopure sample would form (R,R)- or (S,S)-**Agg 3**. After complete transformation of the mixture with an ee of 50% of (R,R)-**PBI**, we found that the solution contained about 50% **Rac-Agg 4** which is formed from 25% of (R,R)-**Agg 1** and 25% (S,S)-**Agg 1**, demonstrating that **Rac-Agg 4** is the thermodynamically stable supramolecular polymer within the racemic mixture. The remaining ~50% of the solution are composed of (R,R)-**Agg 1** which did not nucleate and transform into another species within 6 h of ultrasonication.

We added a sentence ("Notably, the formation of **Rac-Agg 4** was also observed within a mixture of (R,R)- and (S,S)-**PBI** with an enantiomeric excess (ee) of (R,R)-**PBI** of ee = 50% at the same concentration ($c_T = 4.0 \times 10^{-4}$ M, for further details see Supplementary Fig. 7b,c)) on page 9 of the main manuscript and a new figure in the Supplementary Information (Supplementary Figure 7).

What if a mixture of (R,R)-Agg3 and (S,S)-Agg3 is agitated?

Answer: In order to clarify this question, we prepared (R,R)- and (S,S)-**Agg 3** from the respective enantiomerically pure solution and mixed them in a 1:1 ratio which afforded **Con-Agg 3**. In the manuscript we called this procedure "manual preparation" of **Con-Agg 3**. After prolonged sonication of **Con-Agg 3**, the respective mixture transformed into the thermodynamically stable racemic supramolecular polymer **Rac-Agg 4** (the respective absorption spectra are provided in the Supplementary Figure 15). We added a discussion in the manuscript (page 20 and 21) and Supplementary Information (page 19 and 20) and updated the final potential energy surface (Figure 6, main manuscript) which now also shows the transformation pathway from manually prepared **Con-Agg 3** to **Rac-Agg 4**.

Reviewer 2:

This manuscript reports control on supramolecular polymerization of a mixture of enantiopure (R, R)- + (S, S) PBI derivatives leading to the formation of conglomerate and racemic structures depending on specific conditions. This phenomenon has been elucidated with careful AFM analysis, and UV/Vis spectroscopy of the homo-polymers and mixtures while the kinetic and thermodynamic analysis was conducted by spectroscopy studies. Experimental observation has been supported by theoretical studies which together were used to provide a clear picture on this complex supramolecular polymerization process and chirality issues. While chirality issues in supramolecular polymerization has been studied with significant detail in the recent past, most examples are limited to chirality induction and in the recent past chiral luminescence. In contrast the present study reflects a fundamental aspect which, although was known in context of organic crystals, has never been looked at in detail in context of 1D supramolecular polymers as rightly claimed by the authors. The paper has been written very well, it is remarkably free of error and should make a lasting impact in the field. The manuscript is recommended for publication after minor revision as noted below:

(a) It may be relevant to examine the supramolecular assembly of (R, R)- + (S, S) PBIs with stoichiometric imbalance. At least with one ratio other than 1:1 may be informative. Will it still produce a conglomerate and, in that case, will it lead to eventually homochiral assembly following majority rule?

Answer: In Supplementary Figure 7 we now provide studies on solutions with enantiomeric imbalances (50% ee (R,R)-PBI, i.e. solutions containing 75% (R,R)-PBI and 25% (S,S)-PBI). The outcome of this study is that the respective solutions behave similarly as the racemic mixtures: Upon ultrasonication at 298 K, a mixture containing homochiral (R,R)-Agg 2 and remaining (not nucleated) (S,S)-Agg 1 are formed showing (homochiral assembly of the individual components. Likewise, ultrasonication at 308 K yielded a 1:1 mixture of Rac-Agg 4 and (R,R)-Agg 1. A single, homochiral supramolecular polymerization with incorporation of the minority species into the majority aggregate species following the "majority-rules" effect was not observed in these experiments. In the main manuscript we added a sentence on page 7+8 and page 9, respectively, to address these questions.

(b) Do different aggregated states exhibit distinct emission spectra?

Answer: Previous studies on (R,R)-Agg 1–3 (Wehner, M. et al. Supramolecular Polymorphism in One-Dimensional Self-Assembly by Kinetic Pathway Control. J. Am. Chem.

Soc. **141**, 6092-6107 (2019)) showed that **(R,R)-Agg 1** and **(R,R)-Agg 2** exhibit almost identical excimer-type emission spectra. Since **Con-Agg 1** and **Con-Agg 2** are composed of **(R,R)-Agg 1** and **(S,S)-Agg 1** or **(R,R)-Agg 2** and **(S,S)-Agg 2**, respectively, distinct emission spectra of the conglomerates cannot be expected. Together with the fact that the aggregates are almost non-fluorescent (very low fluorescence quantum yields below < 1%) which hampers the accurate investigation of emission spectra, emission spectra are not provided in our study.

(c) Supplementary Figure 4: It is intriguing that the spectra at 295K are particularly broad and become relatively sharp again by lowering the temperature further. What may be the reason?

Answer: We added a short discussion in the Supplementary Information below Supplementary Figure 4 to discuss this issue. We assume that the broadening at 295 K stems from a dynamic equilibrium between monomers and rather dynamic dimers which are not yet fully interlocked in their stable conformation with close π -contacts and fourfold intermolecular hydrogen-bonding.

(d) A recent review article (Chem.Comm., 2020, 56, 6757) on controlled supramolecular assembly may be relevant in context of the discussion in the introduction.

Answer: We included this review article in the introduction (reference 21) as it highlights recent examples of kinetic supramolecular polymerization.

Reviewer 3:

This article reports the discovery of new polymorphs of supramolecular polymers composed of a racemic mixture of perylene bisimide (PBI) as well as the formation pathways of those polymorphs. This also represents the first mechanistic study of one-dimensional supramolecular polymerization of a racemic mixture of (R,R) and (S,S)-PBI. It was determined that a racemic mixture of (R,R) and (S,S)-PBI initially formed homochiral dimer pairs in solution (the conglomerate particles of which are termed Con-Agg 1), but subsequent perturbation either by sonication (purported to induce kinetic self-assembly) or heating and cooling as well as sonication (purported to induce thermodynamic self-assembly) could induce the formation of conglomerate, H-bonded homochiral helices (termed Con-Agg 2) or periodic supramolecular polymers comprised of alternating homochiral pairs (termed Rac-Agg 4), respectively.

Answer: Maybe our formulation “For the investigation of conglomerate versus racemic compound formation (Figure 1a), racemic mixtures of (R,R)- and (S,S)-PBI were subjected to heating – cooling cycles...” was a little bit misleading (see below). Accordingly, we changed that formulation into “For the investigation of conglomerate versus racemic compound formation (Fig. 1a), racemic mixtures of (R,R)- and (S,S)-PBI in MCH/Tol (5:4 v/v) were prepared as described in the methods section.” and provided a more detailed description of the preparation of the racemic mixture and the ultrasonication procedure in the methods section.

It is noteworthy, that both species, **Con-Agg 2** and **Rac-Agg 4**, were prepared according to the same methodology, that is ultrasonication of a solution of the racemic mixture either at 293 – 298 K (**Con-Agg 2**) or at 308 K (**Rac-Agg 4**). In both cases, we started from a hot solution of the racemic mixture in which the ultrasonication finger was dipped followed by sealing of the vial and placing it in a water bath (for the transformation into **Con-Agg 2** the temperature of the water bath was 293 – 298 K, for the transformation into **Rac-Agg 4** the temperature was 308 K).

The formation of these species (Con-Agg 2 and Rac-Agg 4) was determined to proceed via nucleation-elongation mechanisms, as evidenced by sigmoidal kinetic traces. Thermodynamic analysis of the dissociation of Rac-Agg 4, as well as quantum chemical calculations, showed that the formation of Rac-Agg 4 was more thermodynamically favorable than the formation of Con-Agg 2. Furthermore, the authors claim that the concentration-dependent kinetics of formation of Rac-Agg 4 from Con-Agg 1 indicate that Con-Agg 1 is an on-pathway intermediate to the formation of the putative thermodynamic product Rac-Agg 4. This study thus represents a thorough characterization of the thermodynamics of a system capable of supramolecular self-assembly and gives guidelines for pathway control within this system.

The purported structures of each of the novel polymorphs are supported by UV/visabsorption studies, circular dichromism data, and quantum chemical calculations. These calculations also support the categorization of Rac-Agg 4 as the thermodynamically most stable state of the system and Con-Agg 2 as a kinetic product, since it is thermodynamically more stable than Con-Agg 1 but not as stable as Rac-Agg 4. The authors are careful not to overstate the thermodynamic stability of Rac-Agg 4, but occasionally use ambiguous language to describe the relative stability of each polymorph.

For example, in the Discussion section, the authors claim that Rac-Agg 4 is formed under thermodynamic control, and in their qualitative energy surface (Figure 6) place a large energetic barrier between the minima corresponding to Con-Agg 2 and Rac-Agg 4.

Answer: **Con-Agg 2** and **Rac-Agg 4** are kinetically stable species at sufficiently high concentrations and do not spontaneously interconvert into each other at room temperature. Therefore, we drew a large energetic barrier between **Con-Agg 2** and **Rac-Agg 4** that accounts for the kinetic stability of these species. Furthermore, we observed that **Rac-Agg 4** is directly formed from **Con-Agg 1** upon ultrasonication at 308 K. Ultrasonication at 308 K provides enough energy to overcome the kinetic barriers between **Con-Agg 1** and **Con-Agg 2** or the one between **Con-Agg 1** and **Rac-Agg 4**. Since **Rac-Agg 4** is thermodynamically more stable than **Con-Agg 2** (which also means that a solution of **Con-Agg 2** disassembles more easily compared to an equally concentrated solution of **Rac-Agg 4**), **Rac-Agg 4** is formed under these conditions.

Notably, we observe isosbestic points in the UV/vis spectra for the transformation of **Con-Agg 1** into **Rac-Agg 4** (Figure 4b) which suggests that only two species, namely **Con-Agg 1** and **Rac-Agg 4** (no **Con-Agg 2**), are involved in the transformation process. Since ultrasonication at 308 K provides enough energy to overcome the kinetic barriers between **Con-Agg 1** and **Con-Agg 2** or the one between **Con-Agg 1** and **Rac-Agg 4**, this means that the kinetic barrier between **Con-Agg 1** and **Rac-Agg 4** must be lower than the kinetic barrier between **Con-Agg 2** and **Rac-Agg 4** because otherwise the system would transform into **Con-Agg 2** and not directly into **Rac-Agg 4**. Therefore, the relative heights of the kinetic barriers, albeit being drawn only qualitatively, are correctly represented in Figure 6.

Since pathway independence is characteristic of thermodynamic equilibria for dynamic systems, dual entry point analysis experiments would bolster the authors' argument that **Rac-Agg 4** is the most thermodynamically stable species. Specifically, the authors might consider sonication and thermal cycling of **Con-Agg 2** to potentially generate **Rac-Agg 4**. This would determine whether **Con-Agg 2** is a kinetic trap or an off-pathway intermediate en route to the putative thermodynamic product **Rac-Agg 4**.

Answer: As stated above, probably our formulation of heating – cooling cycles for the preparation of the polymorphs was misleading here and therefore we changed it.

Notably, the polymorphs **Con-Agg 2** and **Rac-Agg 4** could only be produced by ultrasonication. We could never produce them by pure heating – cooling cycles, because heating leads to the dissociation of the respective species (for further details see Wehner, M. et al. *Supramolecular Polymorphism in One-Dimensional Self-Assembly by Kinetic Pathway Control*. *J. Am. Chem. Soc.* **141**, 6092-6107 (2019)) into **Con-Agg 1** and monomers (see also Figure 5) and cooling of hot solutions of the racemic mixture always led to the

instantaneous formation of **Con-Agg 1**. Therefore, thermal cycling with or without ultrasonication cannot be used for the transformation of **Con-Agg 2** into **Rac-Agg 4**.

However, we are able to address the reviewer's suggestion to generate **Rac-Agg 4** from **Con-Agg 2** by ultrasonication (see Supplementary Figure 16). Therefore, we sonicated a **Con-Agg 2** solution at 308 K (without any further heating or cooling). Interestingly, after 10 min ultrasonication, **Con-Agg 2** disassembled into **Con-Agg 1**. Afterwards, **Con-Agg 1** was transformed into **Rac-Agg 4** within 310 min and successive equilibration at room temperature.

This experiment proves that

- 1) **Rac-Agg 4** is the thermodynamically stable state and more stable than **Con-Agg 1** and **Con-Agg 2** (since the final outcome of the ultrasonication led to the formation of **Rac-Agg 4**)
- 2) **Rac-Agg 4** is formed from **Con-Agg 1** and not directly from **Con-Agg 2**, meaning that the potential surface with the relative barrier heights provided in Figure 6 is right.

We additionally illustrated the new transformation pathway from **Con-Agg 2** to **Rac-Agg 4** with **Con-Agg 1** as an intermediate in Figure 6 and added the following sentence on page 21: "Additionally, we were able to transform **Con-Agg 2** into **Rac-Agg 4** upon ultrasonication at 308 K (Supplementary Fig. 16) which proceeds via disassembly of **Con-Agg 2** into **Con-Agg 1** (magenta arrow, Fig. 6) followed by the transformation of **Con-Agg 1** into **Rac-Agg 4** (orange arrow, Fig. 6)."

Since **Rac-Agg 4** can also dissociate into **Con-Agg 1** (homochiral dimers), it would also be worth determining whether or not **Rac-Agg 4** can be converted into any other metastable species, particularly **Con-Agg 2**, by slow thermal cycling and prolonged sonication. This would better inform the authors' representation of the system's energetic landscape and elucidate other possible formation pathways for each polymorph.

Answer: All polymorphs need to be prepared by ultrasonication of a solution of **Con-Agg 1** as outlined in the methods section. As stated above, thermal cycling does not work to prepare different polymorphs.

Rac-Agg 4 is the thermodynamically stable state and does not spontaneously transform into any other species over time. Additionally, we checked prolonged sonication of **Rac-Agg 4** at

308 K and did not see any transformation of **Rac-Agg 4** into another species (**Rac-Agg 4** remained stable). Furthermore, as shown in Supplementary Figure 15, prolonged sonication of a manually prepared **Con-Agg 3** mixture at increased temperature of 313 K (6 h sonication) also led to the formation of thermodynamically stable **Rac-Agg 4** and not to the transformation into another species.

We added the following sentence on page 19 of the main manuscript: "**Rac-Agg 4** does not spontaneously transform into any other species at room temperature over time and additional experiments to transform **Rac-Agg 4** into another species upon prolonged (> 2 h) ultrasonication ($T \leq 313$ K) failed, which reflects the high thermodynamic stability of **Rac-Agg 4**."

There are occasional minor grammatical and spelling errors throughout the text, but it did not impact my understanding of the article.

Answer: We corrected these mistakes to the best of our knowledge.

This article demonstrates a generalizable method to map the energy landscape of a dynamic system capable of producing multiple products, which may become a very useful tool to simply represent complicated dynamic systems. This work also uses thermodynamic modeling to elucidate purported kinetic and thermodynamic pathways in supramolecular polymerization, a task which is often overlooked in computational approaches to dynamic combinatorial chemistry. However, the work would be more complete and impactful if the relationship between each of the novel species reported was further elucidated. That is, analysis of pathway-dependence in the synthesis of **Rac-Agg 4**, as well as further thermodynamic characterization of **Con-Agg 2** to determine whether it is a kinetic trap or an off-pathway intermediate would expand the author's claim of "control of self-assembly pathways." For these reasons I recommend publication with major revisions.

Answer: With the additional experiments suggested by the three reviewers we provided further evidence for the thermodynamic stabilities of the four aggregate species that can be obtained from (R,R)- and (S,S)-PBI. Specifically, we clarified that **Con-Agg 2** is a kinetically trapped state and not the thermodynamically stable state, which is **Rac-Agg 4**. Furthermore, we elucidated a new transformation pathway from **Con-Agg 2** mixture to **Rac-Agg 4** with **Con-Agg 1** as an intermediate. Therefore, Figure 6 could be augmented with a magenta arrow which shows that the reverse reaction (reaction from **Con-Agg 2** to **Con-Agg 1**) is also possible as well as the transformation pathway from manually prepared **Con-Agg 3** to **Rac-**

Agg 4. Thus, we were finally able to investigate all transformation pathways between the respective species.

REVIEWERS' COMMENTS

Reviewer #1 (Remarks to the Author):

The authors clearly addressed the comments raised by the reviewers. I am happy to recommend the publication of this manuscript in Nature Communications.

Reviewer #2 (Remarks to the Author):

This is fine revision, authors have addressed all issues pointed out by the reviewers. The revised manuscript is recommended for publication.

Reviewer #3 (Remarks to the Author):

The authors' revised ms has satisfactorily addressed all of my concerns.